# HPF1 Regulates Pol β Efficiency in Nucleosomes via the Modulation of Total Poly(ADP-Ribose) Synthesis

**DOI:** 10.3390/ijms26051794

**Published:** 2025-02-20

**Authors:** Mikhail Kutuzov, Dinara Sayfullina, Ekaterina Belousova, Olga Lavrik

**Affiliations:** Institute of Chemical Biology and Fundamental Medicine (ICBFM) SB RAS, 630090 Novosibirsk, Russia; kutuzov.mm@mail.ru (M.K.); din.sayfullina@gmail.com (D.S.); rina@niboch.nsc.ru (E.B.)

**Keywords:** nucleosome, poly(ADP-ribose)polymerase, PARP1, PARP2, base excision DNA repair, BER, histone PARylation factor 1, HPF1, DNA polymerase β

## Abstract

The maintenance of genome stability and the prevention of genotoxic damage to DNA require immediate DNA repair. In the cell, the repair process is usually preceded by a release of DNA from complexes with chromatin proteins accompanied by nucleosome sliding, relaxing or disassembly. Base excision DNA repair (BER) corrects the most common DNA lesions, which does not disturb the DNA helix dramatically. Notably, small DNA lesions can be repaired in chromatin without global chromatin decompaction. One of the regulatory mechanisms is poly(ADP-ribosyl)ation, leading to the relaxation of the nucleosome. In our work, we demonstrated that recently a discovered protein, HPF1, can modulate the efficiency of one of the key BER stages—DNA synthesis—via the regulation of total poly(ADP-ribosyl)ation. Accordingly, we investigated both short-patch and long-patch DNA synthesis catalyzed by DNA polymerase β (pol β; main polymerase in BER) and showed that HPF1’s influence on the poly(ADP-ribosyl)ation catalyzed by PARP1 and especially by PARP2 results in more efficient DNA synthesis in the case of the short-patch BER pathway in nucleosomes. Additionally, HPF1-dependent poly(ADP-ribosyl)ation was found to positively regulate long-patch BER.

## 1. Introduction

Lesions constitutively appear in the DNA of living organisms. When unrepaired, DNA damage can lead to mutagenesis, neurodegenerative disorders and cancers [1]. In higher eukaryotes, there are several DNA repair systems, which are specific to different types of DNA damage. Base excision DNA repair (BER) is required for the repair of lesions that appear under oxidative stress and during other genotoxic processes. These types of damage do not significantly disturb the structure of the DNA double helix [1]. Multiple studies have been devoted to BER; thus, to date, the main stages of this process have been well characterized in the context of naked DNA model structures [2,3,4,5,6,7]. Nonetheless, some factors, such as chromatin compaction and chromatin reorganization, can affect DNA repair in general and the efficiency of each BER enzyme in particular.

The basic level of DNA compaction is the nucleosome (nucleosome core particle, NCP). It is composed from 147 bp DNA wrapped around a histone octamer that has a pseudo dyad, whose axis passes through the central base pair in the DNA sequence [8]. Such organization implies the nonequivalence of nucleotides in the DNA duplex in terms of protein accessibility; the latter is determined by two main factors: the position of a nucleotide in relation to the dyad and the orientation of a nucleobase in relation to the histone core [9,10]. Firstly, the nucleobase with the most hindered access is located at a distance of a few nucleotides from the dyad. Secondly, a nucleobase can be rotationally oriented towards the NCP core or in the opposite direction, denoted as “in” and “out”, respectively. The model of NCP organization is the simplest approximation of a proper chromatin model where an NCP is the structural unit and reflects the influence of structural organization on the functioning of DNA-dependent processes. Undoubtedly, it is necessary to take into account the effect of histones and DNA modifications’ influence on NCP stability, as well as differences in the mobility of NCPs in a chromatin context in vivo [10]. Nevertheless, the NCP as a model is useful in molecular biological and biochemical studies. Researchers frequently use the DNA sequences designed in Widom’s lab, which are known as clones 601 and 603. These DNA sequences have high affinity for histones during NCP formation and are characterized by only one positioning toward the NCP core.

Because the organization of DNA into an NCP makes adjustments in DNA-dependent processes, it also affects DNA repair in general and BER in particular. Research data testify to the repression of BER in an NCP compared with naked DNA [9,11,12,13,14]. This phenomenon depends on several parameters: the rotational orientation of a damaged nucleotide in relation to the NCP core, the distance between the damaged nucleotide and the NCP dyad and the DNA lesion context used in a model DNA sequence. Nonetheless, specific features of the enzyme catalytic stage are also important. For example, proteins that do not require a dramatic distortion of DNA geometry in an NCP face fewer obstacles to their functioning [15].

In general, BER is an important repair mechanism owing to its activity throughout the cell cycle. BER includes the recognition step involving the excision of a damaged base and the incision of the sugar–phosphate backbone, with subsequent gap processing and DNA ligation. One of critical participants in this process is DNA polymerase β (pol β), which possesses DNA polymerase and dRp-lyase activities for the implementation of gap filling before ligation [2,5,11,12]. The most intriguing field of BER research right now is the regulation of this multistage process and investigation into backup pathways under conditions where the canonical pathways of BER cannot be realized. In this regard, there is a significant mechanism of regulation: poly(ADP)ribosylation [3]. Poly(ADP-ribose) (PAR) is a signaling molecule that is important in the formation in the DNA damage response, including BER regulation. PARylation is mostly catalyzed by poly(ADP-ribose)polymerases 1 and 2 (PARP1 and PARP2), which synthesize long-branched polymers of ADP-ribose [16,17]. In response to DNA damage, both PARPs are activated and synthesize PAR covalently attached to an acceptor molecule, which could be BER enzymes in general and pol β in particular, or scaffold proteins or other molecules of PARPs. PARylation performs some important functions, such as signaling for recruiting PAR-binding proteins to a DNA damage site and the modulation of properties of a protein being modified [18]. Moreover, PARylation can lead to the dissociation of a complex of the modified protein with DNA [19,20,21].

It should be noted that the basic activity differs between PARP1 and PARP2 and depends not only on the type of DNA substrate but also on additional factors [16,20]. For example, a recently discovered protein, a histone PARylation factor called HPF1, is a modulator of PARP1 and PARP2 activities [22]. It complements the active site of these enzymes, thereby modifying their activity and specificity and leading to the efficient PARylation of core histones. This modulation is considered one of the regulatory mechanisms because the PARylation of core histones can lead to NCP destabilization, loosening core histones or, probably, uncoiling.

Biochemical studies on the HPF1-mediated modulation of the activity of PARP1 and PARP2 have revealed significant changes in the substrate specificity and kinetic parameters of these enzymes [23,24]. In particular, in ref [24], it was demonstrated that HPF1 distinctly stimulates autoPARylation and histone heteromodification by both PARPs at low NAD^+^ concentrations, at which mono(ADP-rybosil)ation prevails. This evidence indicates that a major function of HPF1 is probably performed in the early stages of PARylation or during NAD^+^ depletion. Moreover, the most significant HPF1-induced stimulation of histone modification has been demonstrated in the presence of an NCP with gapped DNA mimicking the central BER intermediate [25].

Taken together, these data imply a kind of specific regulatory role of PARP2 in DNA repair in the context of chromatin. Accordingly, this notion encourages investigation into the influence of PARP1 and PARP2 on BER in the presence of HPF1. Here, we analyzed the influence of PARP1- or PARP2-mediated PARylation in the presence of HPF1 on the polymerase efficiency of pol β in terms of the catalysis of the incorporation of one nucleotide (under single-nucleotide gap-filling conditions) or a few nucleotides (under conditions of DNA strand-displacement synthesis) in a gap-containing NCP.

## 2. Results and Discussion

In our study, we focused on the regulation of pol β polymerase activity in the NCP context by using gapped DNA substrates mimicking the BER intermediate. To be precise, this work is devoted to research on the effects of HPF1-modulated total PARylation on pol β activity. During the BER process, pol β can either incorporate one nucleotide filling the gap or catalyze the incorporation of up to a few nucleotides with DNA strand displacement, depending on the BER pathway. In this project, we examined both versions of event: single-nucleotide gap filling and DNA strand displacement.

For this purpose, we designed an NCP based on Widom’s clone 603 DNA sequence with a single-nucleotide gap. We chose the position of the one-window gap in such a way that it yields the “out” orientation of the first inserted dNMP. Because the nucleosome organization of DNA can regulate patch size during BER according to the rotational orientation of nucleotides [12], an additional criterion was the accessibility of approximately three nucleotides downstream of the gap position. This condition is required for strand-displacement synthesis catalyzed by pol β. Moreover, this gap is located at the 21st nucleotide position from the 5′ end: quite distant from the dyad. This combination ensures access to DNA-binding proteins, thereby making the monitoring of dynamics of biochemical substrate transformation possible. In addition, this position is also far from the end of the DNA entry–exit site of the NCP and therefore is in a quite dynamic region [26]. Additionally, the distance of more than 20 nt from the blunt ends of the model NCP helps to avoid the characteristics of clustered DNA damage.

The activity of pol β is known to be regulated by PARP1 and PARP2 on an NCP substrate, as well as on naked DNA [9,27]. Both PARPs suppress the nucleotide transferase activity of pol β to different degrees; however, PARylation restores the enzymatic activity [9,27].

First, we performed the experiments under single-nucleotide gap-filling conditions. In this case, only dTTP was added to the reaction mixture. At first, we titrated pol β to select the concentration ensuring a nucleotide incorporation efficiency of 40–70% (Appendix A). This range provided the opportunity to evaluate the effect of a PARP on DNA synthesis. Therefore, further experiments were conducted in the presence of 0.3 nM pol β with naked DNA or 60 nM pol β with an NCP. These differences could be explained by the existence of nucleosome structure complexity that suppresses substrate processing by the DNA polymerase activity of pol β.

According to literature data, PARP proteins significantly reduce pol β activity because of competition for the substrate [20,28]. We assessed such an effect of PARP1 and PARP2 by using both naked DNA and the NCP. The quantified data are presented in Figure 1 and Appendix A.

According to these results and keeping in mind that the affinity of both PARPs for gapped DNA and gapped NCP structures is in the ranges of 30–60 and 10–15 nM, respectively [20], in further experiments with naked DNA and the NCP, we decided to employ 30 and 100 nM PARPs, respectively, to be able to evaluate the PARylation effect and the influence of HPF1 supplementation. Here, in panels (a) and (b), for both PARP1 and PARP2, the *p*-values for the differences between bars 1 and 3 are ≤0.001.

Because HPF1 is a factor modulating the specificity and PARylation activity of PARP1 and PARP2, we estimated pol β-catalyzed single-nucleotide gap filling under PARylation and determined the influence of HPF1 on this process (Figure 2). The raw data are presented in Appendix A.

The obtained results clearly indicate that both PARPs suppress the nucleotide incorporation catalyzed by pol β on naked DNA or the NCP (Figure 1 and Figure 2). This finding is an agreement with the literature data, where researchers have demonstrated the binding of PARP1 and PARP2 to DNA mimicking BER intermediates [16,20,27,28,29,30,31]. The slight difference between PARP1 and PARP2 in the effect on pol β could be explained by a difference between PARP1 and PARP2 in the affinity for DNA-containing substrates. Definitely, the overall affinity of PARP1 for different forms of DNA mimicking the DNA intermediates of BER, either naked DNA or DNA in the nucleosomal context, is stronger compared with PARP2, thus possibly resulting in more effective competition with pol β and the consequent stronger effect on dNMP incorporation.

PARylation attenuated the impact of each PARP on pol β activity, and this finding was more pronounced with PARP1 in the presence of DNA and of the NCP (bars 2–6 in panel (a) and (b) of Figure 2; for PARP1, the *p*-value of the difference between bars 2 and 6 in panels (a) was ≤0.01, and that of the difference in panels (b) was ≤0.05). The suppressive effect diminished in accordance with the efficiency of the formation of various PARs by PARP1 or PARP2. It is known that PARP1 is more active; therefore, PARylation by PARP1 attenuates suppression more strongly compared with PARP2 [28].

Notable changes were caused by the presence of HPF1 (Figure 2). It should be pointed out that the addition of HPF1 without NAD^+^ to the reactions with PARP1 did not affect the catalytic activity of pol β (in Figure 2, compare bars 2 and 7; Appendix A; in Figure 2, for PARP1, the *p*-value of the difference between bars 2 and 7 in panels (a) was ≤0.01, and that of the difference in panels (b) was ≤0.05). On the contrary, the supplementation of PARP2 with HPF1 led to a slight restoration of the dTMP incorporation level (in Figure 2, compare bars 2 and 7; for PARP2, the *p*-value of the difference between bars 2 and 7 in panels (a) was >0.05, and that of the difference in panels (b) was ≤0.05). Nevertheless, the overall impact of HPF1 manifested itself clearly under PARylation (compare the trend of bars 3–6 with the one of bars 8–11 in panels (a) and (b) of Figure 2).

In this regard, the effect depends on substrate complexity. Initially, when we started with the lowest concentrations of NAD^+^, the influence of PARP1 on pol β activity on naked DNA under PARylation was not dramatically changed by the presence of HPF1. Earlier, it has been reported that supplementation with HPF1 stimulates PARP1’s autoPARylation in the presence of a gapped NCP; therefore, this phenomenon could induce the dissociation of the initial PARP1/gap-NCP complex and allow pol β to access a DNA substrate [25]. In this context, the strengthening of the PARP1 effect on pol β activity could be a cumulative consequence of the targeting of alternative amino acids by HPF1 in the initial stages. To evaluate the effect of HPF1 in the case of PARP2 on the activity of pol β, a higher concentration of PARP2 or of PARP2 supplemented with HPF1 in the same molar ratio was tested next (Appendix A). Under these conditions, a greater effect was detected, confirming the tendency observed in the previous experiments.

If we examine results of the experiment on the NCP, first, it should be noted that they cannot be compared directly with the experiments on DNA because of the substantial dissimilarity of the reaction conditions (compare panels (a) and (b) in Figure 2). Nevertheless, the main trend is the same: definitely, both PARPs suppress pol β activity, which gets restored under PARylation conditions, as observed above with naked DNA (the trends of bars 2–6 in panel (b) of Figure 2).

Considerable differences between the experiments on naked DNA and on the NCP appeared in the presence of HPF1 under PARylation. Here, the activity of pol β in the presence of PARP/HPF1 complexes was higher than that without HPF1, especially for PARP2 (compare bars 5–6 and 10–11 in panel (b) of Figure 2). Moreover, the final magnitude of nucleotide incorporation was found to be increased compared with the control, involving pol β alone (compare bar 6 with bar 11 for PARP1, where the *p*-value was ≤0.05, and bars 5 and 6 with bars 10 and 11 for PARP2, where the *p*-value was ≤0.05, in panel (b) of Figure 2). This phenomenon is probably a consequence of a higher autoPARylation level of PARPs and the subsequent switching of activity to alternative modification targets that are histones in the presence of HPF1 [24]. Histones’ PARylation relaxes the NCP structure, thus facilitating access to DNA for pol β and enabling more efficient nucleotide incorporation. On the other hand, PARP/HPF1 complex formation may cause a shift in the competition between a PARP and pol β for the substrate in favor of DNA polymerase. It is also noteworthy that the effect of PARP2 was more pronounced, indicating more efficient stimulation of PARP2 activity by HPF1. This finding might reflect the differences in the modulation of PARP1 and PARP2 activity by the presence of HPF1.

Earlier studies described a stimulatory effect of HPF1 on different stages during PAR synthesis [24,25]. These fundings are especially important in the context of the complexity of the NCP structure and the stability of the core particle. To clarify this issue, we analyzed the kinetics of PAR accumulation catalyzed by either PARP1 or PARP2. The data are presented in Figure 3. It should be noted that a direct comparison of the efficacy of PAR accumulation between the experimental results presented in different panels of Figure 3 is impossible due to varied initial reaction conditions. Therefore, we describe the differences obtained in the parallel experiments only with HPF1 supplementation.

The supplementation of PARPs with HPF1 led to a decrease in the amount of PAR in the experiments with naked DNA. Possibly, this outcome is a consequence of the PARylation target specificity of each PARP/HPF1 complex compared with a PARP alone; this phenomenon leads to the modification of serine exclusively instead of glutamate, aspartate and other amino acid residues. Activation by more complicated DNA substrate, the NCP, results in an increase in PAR accumulation in the reaction with PARP2 and HPF1 (Figure 3). Nevertheless, the activity of PARP1 toward the same substrate was overall suppressed by HPF1 under these conditions. According to the Michaelis–Menten classic scheme as the first approximation, the possible reduction in k_cat,adj_ during the supplementation of PARP1 with HPF1 during PARylation could explain the latter finding. Here, the consumption of NAD^+^ molecules owing to the DNA-independent hydrolytic activity of the PARP/HPF1 complex cannot be discerned in the overall results; thus, we can only speculate about k_cat,adj_ [32]. Considering a prior study estimating the impact of HPF1 on the initiation phase of PAR synthesis, particularly for PARP1, it could be inferred that the PARP1/HPF1 complex undergoes stronger binding to NAD^+^ [25]. This hypothesis suggests that the presence of HPF1 may potentially inversely affect the PAR synthesis rate at different NAD^+^ levels and that this effect depends on which step is rate-limiting: NAD^+^ binding or catalysis.

Previously, in Ref. [25], more efficient PARylation of histones of an NCP containing gapped DNA by PARP2 was shown in comparison to PARP1. In that paper, the reactions were carried out in the presence of low NAD^+^ concentrations, mimicking the initiation stage of PAR synthesis, i.e., when the NAD^+^ binding can determine the effectiveness of the entire process of PAR synthesis. Here, we conducted experiments in the presence of an excess of NAD^+^, which should ensure the predominance of the synthesis of longer PAR by PARP enzymes. Under these conditions, the rate of PAR synthesis is primarily determined by the catalytic step, not binding ones. This observation has the following possible implications: Indeed, in the initial part of the kinetic data, PARP1 activity is influenced by HPF1, whereas later, the growth of the reaction curve becomes slower (Figure 3). Simultaneously, HPF1 stimulates PARP2 activity on the NCP (Figure 3), implying a higher PAR synthesis rate for this protein in the presence of HPF1. In this case, the effective PAR synthesis catalyzed by PARP2 in the presence of HPF1 could be a consequence of ongoing PARylation of histones [25]. This evidence, alongside the data on PARPs’ target switching in the presence of HPF1, points to a specific role of PARP2 in the regulation of DNA compaction in the NCP during BER [25].

When DNA cannot be ligated after the incorporation of a single nucleotide, a DNA polymerase continues DNA synthesis in a strand-displacement manner for a few nucleotides. To reproduce such conditions in vitro, we supplemented the reaction with all four dNTPs. It should be noted that strand-displacement synthesis within a nucleosome is highly dependent on the initial damage location, and its efficiency could vary by 2–15 times relative to mononucleotide incorporation [12]. Therefore, in our case, we used the starting position of a gap at the beginning of the DNA superhelix to support accessibility in a maximal number of steps during DNA polymerase synthesis.

At first, we titrated the pol β concentration to detect the incorporation of a few nucleotides. Basically, the 3′ end was extended by an additional two–three nucleotides. In this case, almost exhaustive gap filling should take place. Accordingly, we used 3.0 nM pol β for the experiments with DNA. To ensure comparable efficiency in the experiment on the NCP, 1.0 µM pol β was utilized; additionally, the NCP structure suppresses DNA polymerase activity. The raw data are presented in Appendix A.

Then, we found the concentrations of PARP1 and PARP2 when they significantly reduced the DNA synthesis. To this end, we titrated PARP1 and PARP2 (Appendix A). The quantified data are presented in Figure 4.

According to the results, we selected the 100 nM concentration of each PARP for the experiments on naked DNA. Here, in panels (a), the *p*-value of the difference between bars 1 and 4 for PARP1 was ≤0.001, and that of the difference between bars 1 and 8 for PARP2 was ≤0.001. The PARP concentrations were equal to enable a direct comparison of PAR synthetic activity and the effect of HPF1 (Appendix A). In the experiments with the NCP, we used 1.0 µM PARP1 or 300 nM PARP2 (Appendix A). Here, in panels (b), the *p*-value of the difference between bars 1 and 5 for PARP1 was ≤0.01, and that of the difference between bars 1 and 9 for PARP2 was ≤0.001.

Figure 5 presents the quantification of dNMP incorporation by pol β. The strong inhibitory effect of PARP1 on both first-filling nucleotide incorporation and DNA strand-displacement synthesis catalyzed by pol β on naked DNA was weaker during PARylation (Figure 5 panel (a); for PARP1, the *p*-value of the difference between bars 1 and 2 was >0.001, and that of the difference between bars 2 and 5 was ≤0.001). The addition of HPF1 to the reaction mixtures gradually attenuated the effect of PARylation (Figure 5, panel (a); for PARP1, the *p*-value of the difference between bars 5 and 9 was ≤0.05). Alternatively, in accordance with its weaker DNA affinity, PARP2 suppresses pol β to a lesser extent than PARP1 (Figure 5, panel (b); for PARP2, the *p*-value of the differences between bars 1 and 2 was ≤0.001). The inhibitory effect did not change with the increase in the concentration of NAD^+^ in the presence of PARylation (Figure 5, panel (b); for PARP2, the *p*-value of the difference between bars 2 and 5 was >0.05); however, the addition of HPF1 substantially promoted the first step of DNA synthesis, i.e., gap-filling, at a high NAD^+^ concentration (Figure 5, panel (b); for PARP2, the *p*-value of the difference between bars 5 and 9 was ≤0.01). These data are consistent with the lower efficiency of PAR synthesis by PARP2 compared with PARP1. Earlier, we have observed a similar effect with short model DNA duplexes without HPF1 [28].

After the switch to the complex substrate, i.e., the NCP, the less obvious effect of PARPs or of PARylation on pol β activity was evident. We failed to find any combination of PARP type and NAD^+^ concentration to suppress substrate utilization in comparison to the control samples. Consequently, we analyzed only the proportion of first-filling nucleotide incorporation and DNA strand-displacement synthesis efficiency. Here, the addition of either PARP1 or PARP2 led to negligible suppression of DNA strand displacement (in Figure 5, the *p*-values of the differences between bars 1 and 2 were ≤0.05 for PARP1 (panel (c)) and PARP2 (panel (d))), thereby resulting in a slight shift in the proportion, which was unchanged under PARylation conditions. It is remarkable that both PARP1 and PARP2, when supplemented with HPF1 under PARylation conditions, show a shift in the proportion in favor of DNA strand-displacement synthesis catalyzed by pol β (in Figure 5, the *p*-values of the differences between bars 5 and 9 were ≤0.01 for PARP1 (panel (c)) and PARP2 (panel (d))). Such an effect of HPF1 can give an opportunity to pol β to carry out DNA strand-displacement synthesis in case of obtaining access to the 3′ end of DNA; this access that can be a consequence of NCP relaxation in the presence of the histone PARylation factor. Therefore, after a comparison with previously obtained data [12], we can propose that the strand-displacement synthesis catalyzed by pol β on the NCP depends not only on a DNA lesion’s position but also on HPF1 presence under PARylation conditions.

Due to the difference in PARP1 and PARP2 activity, we clarified the contribution of PARylation by measuring the rate of PAR formation. Because the most dramatic change in pol β activity caused by the presence of a PARP and in subsequent PARylation occurred in the presence of 100 µM NAD^+^ (compare bars 5 with 9 in Figure 5), we conducted kinetic experiments by using this concentration of NAD^+^. The kinetics of PAR formation are presented in Figure 6.

In the presence of PARP1, we observed a similar effect to the conditions of single-nucleotide gap filling only. Over time, the presence of HPF1 led to a twofold decrease in PAR formation efficiency both with DNA and with the NCP. It should be kept in mind that HPF1 did not change the effect of PARP1-driven PARylation on pol β activity toward DNA; however, HPF1 had a positive effect on the reaction with the NCP because of core relaxation (Figure 5). We propose a similar regulatory mechanism here. Due to decrease in PARP1 activity with DNA in the presence of HPF1, the suppressive effect of PARP1 on the activity of pol β under PARylation conditions persisted longer. By contrast, during the interaction with the NCP, the HPF1-mediated switching of PARylation to histones became critical, causing NCP structure relaxation and a subsequent more efficient interaction of pol β with DNA in terms of DNA synthesis.

In the case of PARP2, the effect differs from the single-nucleotide gap-filling conditions, and we propose the same regulatory mechanism as in the case of PARP1. We detected an enhancement in PAR formation promoted by HPF1 in the first kinetic data of the reaction under strand-displacement conditions. It should be pointed out that under the strand-displacement reaction conditions, pol β switches to processive DNA synthesis. Thus, the initial histone PARylation affords DNA accessibility to polymerase, which is in competition with PARP2/HPF1. A consequence is the PAR formation decrease and the simultaneous increase in the yield of the strand-displacement reaction.

In the case of strand-displacement conditions, the difference in the effect size between PARP1 and PARP2 is due to the weaker overall affinity of PARP2 for DNA and for the NCP, except for efficient and specific interaction with gapped DNA [25]. A higher protein concentration should be used in experiments to see a significant effect on catalytic activity. On the other hand, when the reaction is saturated with PARP2, the stimulatory effect of HPF1 is not so pronounced.

## 3. Materials and Methods

### 3.1. Materials

The following reagents and materials were used: recombinant human APE1, recombinant rat DNA polymerase β, pol β, recombinant human PARP1, recombinant murine PARP2 and recombinant human HPF1; histones H2A, H2B, H3 and H4 from *Gallus gallus* were prepared and purified as described in detail previously [24,33,34,35,36]. Recombinant Taq DNA polymerase and *Escherichia coli* uracil-DNA glycosylase (UDG) were kindly provided by Prof. Svetlana Khodyreva (ICBFM SB RAS). DNA primers for the synthesis of 147 bp DNA of Widom’s clone 603 were obtained from the Laboratory of Biomedical Chemistry at ICBFM SB RAS (Novosibirsk, Russia). A plasmid with Widom’s clone 603 sequence pGEM-3z/603 is a gift from V. Studitsky (Addgene plasmid #26658; http://n2t.net/addgene:26658 (accessed on 19 January 2025); RRID: Addgene_26658). Most of the reagents used in the study were purchased from Sigma (Missouri, St. Louis, MO, USA), whereas bromophenol blue and xylene cyanol were obtained from Fluka (Buchs, Switzerland).

DNA amplification products, NCP assembly products and DNA repair synthesis products were visualized after separation in polyacrylamide gel by means of a Typhoon FLA 9500 system (GE Healthcare Life Science, Waukesha, WI, USA)

### 3.2. Methods

#### 3.2.1. Preparation of [^32^P]-NAD^+^

The 32P-labeled NAD^+^ was synthesized enzymatically from [α-^32^P]-ATP (with specific activity of 3000 Ci/mmol; synthesized in the Laboratory of Biotechnology, ICBFM, Novosibirsk, Russia). In particular, 40 μL of the reaction mixture consisted of 300 μM ATP, 10 MBq of α-[^32^P]-ATP, 12 mM MgCl_2_, 2 mM β-nicotinamide mononucleotide and 3.75 mg/mL nicotinamide nucleotide adenylyltransferase (NNAT) in 25 mM Tris-HCl pH 7.5 and was incubated at 37 °C for 1 h. Then, 50 μg of NNAT was added, and the mixture was incubated for an additional 30 min at 37 °C followed by heating to 65 °C for 10 min for enzyme inactivation. After the removal of denatured protein by centrifugation, the solution was used as the reactant.

#### 3.2.2. Preparation of Naked DNA and of NCP

The amplification of double-stranded 5′-FAM-DNA and subsequent NCP reconstitution were performed according to Ref. [37]. The precision NCP-positioning sequence of Widom’s clone 603 DNA served as the template. The DNA was prepared by PCR from the pGEM-3z/603 plasmid vector (Addgene, Watertown, MA, USA). PCR was carried out with the following oligonucleotide primers: Forward, FAM-5′-ACCCCAGGGACTTGAAGTAA[dU]AAGGACGGAGGG-3′; Reverse, 5′-CCCAGTTCGCGCGCCCACC-3′. The purity of each DNA sample was estimated as the homogeneity in electrophoretic mobility on 10% polyacrylamide gel under non-denaturing conditions. The electrophoretic analysis of the preliminary and preparative assembly of the NCP is presented in Appendix A.

#### 3.2.3. Preparing Substrates for DNA Polymerase β

Gap-containing DNA substrates either as naked or NCP-associated DNA, i.e., gap-DNA and gap-NCP, respectively, were obtained enzymatically via the generation of an apurinic/apyrimidinic site, i.e., an AP site, followed by its cleavage. The reaction mixture consisted of 50 nM dU-containing DNA or NCP, UDG (1 activity unit per 0.5 pmol of DNA), APE1 (0.03 μM or 3 μM for DNA and the NCP, respectively) and standard buffer components (50 mM Tris-HCl at pH 8.0, 40 mM NaCl, 5 mM MgCl_2_, 0.1 mg/mL BSA and 1 mM DTT) and was incubated for 30 min at 37 °C. Gap-containing substrates were subjected directly to the following reactions.

Specific conditions for AP site generation and cleavage were found beforehand. The extent of the dU-to-AP site conversion was controlled by alkaline hydrolysis after the addition of 0.1 M NaOH and 1 min incubation at 37 °C, followed by heating for 5 min to 97 °C. The extent of AP site cleavage was controlled by incubation with 20 mM EDTA and 40 mM methoxyamine for 30 min on ice, followed by the addition of an equal volume of formamide and heating for 5 min to 97 °C. In both cases, after heating, separation by electrophoresis on 10% polyacrylamide gel was performed under denaturing conditions.

#### 3.2.4. DNA Synthesis by DNA Polymerase β in Presence of PARP1 and PARP2

The DNA synthesis catalyzed by pol β was implemented in a 10 μL reaction mixture composed of the standard buffer components, 10 nM gap-DNA or gap-NCP and 100 μM dNTP (either 100 μM dTTP in the case of single-nucleotide gap-filling conditions or 100 μM of each dTTP, dCTP, dGTP and dATP in the case of DNA strand-displacement conditions). The concentration of pol β was varied from 0.03 nM to 3 nM for gap-DNA under both single-nucleotide gap-filling and DNA strand-displacement conditions. For gap-NCP, the concentration of pol β varied from 10 nM to 1 μM under single-nucleotide gap-filling conditions and from 30 nM to 3 μM under DNA strand-displacement conditions.

To evaluate pol β efficiency in the presence of a PARP in terms of single-nucleotide gap filling, the selection of the DNA polymerase concentration was based on the reaction yield: 60–70% substrate incorporation within 10 min. This concentration was found to be 0.3 nM for naked DNA and 60 nM for the NCP. Under DNA strand-displacement conditions, the DNA polymerase concentration was selected such that three nucleotides were incorporated. This concentration was found to be 3 nM for naked DNA and 1 μM for the NCP.

The concentrations of both PARP1 and PARP2 were varied from 10 to 300 nM for gap-DNA under both single-nucleotide gap-filling and DNA strand-displacement conditions. In the case of the substrate gap-NCP under single-nucleotide gap-filling conditions, the concentration of each PARP was varied from 10 to 300 nM; under DNA strand-displacement conditions, PARP1 was used in the concentration range from 30 nM to 1 μM, whereas PARP2 from 10 nM to 1 μM. The reaction was initiated by the addition of either gap-DNA or gap-NCP, incubated for 10 min at 37 °C and stopped by adding an equal volume of sample buffer (20 mM EDTA in formamide), followed by the heating for 5 min at 97 °C in case of single-nucleotide gap-filling. To implement the DNA strand-displacement conditions, the reaction was stopped by the addition of 0.001 U of proteinase K and EDTA to the final concentration of 10 mM with incubation for 30 min at 37 °C, followed by the addition of an equal volume of formamide and heating for 5 min at 97 °C.

Reaction products were separated by electrophoresis in denaturing 15% polyacrylamide gel supplemented with 7 M urea, 50 mM Tris, 50 mM boric acid and 1 mM EDTA (pH 8.0). Electrophoresis was performed on 15 × 15 × 0.1 cm plates at a voltage of 30 V/cm and stabilization of voltage. The gels were visualized by means of the Typhoon FLA 9500 scanner (GE Healthcare Life Science, Waukesha, WI, USA), and the yield of the reaction products was analyzed in Quantity One 4.6.6 software basic trial version (Bio-Rad, Hercules, CA, USA). The quantitative data were analyzed in Microsoft Excel 2010 and are presented in histograms as means ± SDs. Additionally, statistical tests based on Student’s *t*-test were performed, and the *p*-values were added in the corresponding parts of the main text.

#### 3.2.5. Influence of HPF1 on PARP-Affected Activity of Pol β

On the basis of the experimental results described in Section 3.2.4, the following concentrations of PARPs were chosen for further assays: for implementing single-nucleotide gap-filling conditions, it was 30 nM PARP1 or PARP2 for naked DNA and 100 nM PARP1 or PARP2 for the NCP; for DNA strand-displacement conditions, both PARPs were used at the concentration of 100 nM for naked DNA, whereas for gap-NCP, PARP1 was applied at 1000 nM or PARP2 at 300 nM.

To study the impact of HPF1, DNA synthesis catalyzed by pol β was carried out as described above with HPF1 supplementation in a 2:1 molar ratio with respect to PARP proteins and with NAD^+^ in the concentration range from 0.1 to 100 μM [25]. The reactions were initiated by the simultaneous addition of NAD^+^ and either gap-DNA or gap-NCP to the protein solution. The reaction mixtures were incubated, and the products were analyzed as described in Section 3.2.4.

For a closer examination, we investigated the impact of HPF1 on the PARP1- and PARP2-mediated regulation of DNA strand-displacement synthesis during PARylation in the presence of 100 μM NAD^+^ only with NCP-associated DNA by using a wide range of PARP concentrations. In this case, PARP1 was used at concentrations from 30 nM to 1 μM, and PARP2, from 2 to 300 nM; HPF1 was applied in a 2:1 molar ratio with respect to PARP protein. The reaction mixtures were incubated, and the products were analyzed as described above.

#### 3.2.6. Kinetic Assay of PARP Activity in the Presence of HPF1

The reactions were carried out in a 40 μL mixture composed of the standard buffer components with the addition of either 5 nM gap-DNA or 10 nM gap-NCP; 1 μM pol β; 100 μM each of dTTP, dCTP, dGTP and dATP; 100 μM NAD^+^ and 0.5 μM [^32^P]-NAD^+^; and 0.3 μM PARP1 or 10 nM PARP2.

The reactions were allowed to proceed either in the absence of HPF1 or with a molar ratio of HPF1 to a PARP protein of 2:1. The mixtures were incubated at 37 °C after the simultaneous addition of NAD^+^ and either gap-DNA or gap-NCP to the protein solution, and the reactions were stopped by dropwise placement of a 5 μL aliquot of the reaction mixture on Whatman 1 paper filters pre-impregnated with trichloroacetic acid (TCA) 30, 60, 120, 180, 300, 420 and 600 s after the start.

PAR, including PAR attached to proteins, was precipitated on filters in the presence of TCA. To remove unreacted NAD^+^, the filters were washed once with 10% TCA for 15 min, then 4 times for 15 min with 5% TCA, and once with 90% ethanol before drying. The amount of synthesized [^32^P]-ADP-ribose was determined by using the Typhoon imaging system (GE Healthcare Life Sciences, Waukesha, WI, USA) and Quantity One 4.6.6 software basic trial version (Bio-Rad). The quantitative data were analyzed in Microsoft Excel 2010 according to the exponential equation and are presented in the figures along with the coefficient of determination *R*^2^.

## 4. Conclusions

The compaction of DNA and repair of DNA lesions are processes that interfere with each other and therefore require precise regulation. One mechanism involved in such regulation is PARylation, which was discovered as a biochemical reaction associated with DNA repair [38]. Multiple studies and especially recent publications indicate that PARylation is an inter-process mechanism of regulation of nucleoprotein complex functioning, including the regulation of DNA compaction [39,40].

Early studies showed a regulatory role of PARP1 and PARP2 in BER [3,16,19,41]. Here, we present a study on the potential involvement of PARP1 and PARP2 in the regulation of single- or several-nucleotide incorporation during BER on an NCP using pol β as the example. For the study, we selected DNA substrate mimicking a BER gap intermediate with the starting position of a gap at the beginning of the DNA superhelix to support accessibility in a maximal number of steps during DNA polymerase synthesis. To avoid cluster damage, regarding the gap and the lesions at DNA ends in the current work, the gap was far from the ends of the DNA.

The DNA polymerase activity of pol β is suppressed by both PARPs, especially when DNA synthesis is accompanied by strand displacement. Basically, the interaction of PARPs with damaged DNA impedes the DNA polymerase activity, whereas PARylation-including the autoPARylation of PARPs leads to their dissociation out of the DNA breaks. Thus, in this context, PARylation is required for the attenuation of DNA polymerase suppression caused by PARPs. The NCP is also an obstacle to successful BER realization owing to substrate complexity. In the cell, there are several mechanisms for overcoming this problem. One of them is the post-translational modification of histones. In terms of PARylation, earlier, H1 has been regarded as a major histone accepting PAR; therefore, it has been supposed that PARylation can affect only a chromatin organization level higher than the NCP [42]. A recently discovered ability of PARP1 and PARP2 to switch PARylation to core histones in the presence of HPF1 radically changed the understanding of the role of PARPs and PARylation in chromatin compaction [43].

Here, we found that under single-nucleotide incorporation conditions, in a gap-containing NCP, both PARP1 and PARP2 impede the activity of pol β. On the other hand, in the presence of PARylation, HPF1 positively affects the kinetics of PAR synthesis catalyzed by PARP2, as well as the total level of PAR formation. Thus, according to our data, PARP2 supplemented with HPF1 is a more promising regulator of pol β dNMP transferase activity in the single-nucleotide BER pathway than PARP1. During strand-displacement synthesis involving the gap-containing NCP, the two PARPs had similar effects on the activity of pol β. Additionally, the shape of the kinetic curves of PAR formation by PARP1 and PARP2 was consistent with the pattern observed with naked DNA and the NCP. Thus, the findings suggest that the combination of PARP2 with HPF1 is more relevant to the effective regulation of single-nucleotide incorporation or strand-displacement synthesis in BER on a gapped NCP than either PARP1 alone or PARP1 with HPF1.

Summarizing the results on pol β activity and PAR formation, we propose a combined mode of the regulation of DNA repair and NCP compaction via PARylation in the presence of HPF1. The overall effect may be supported by dual arguments. In particular, the association of PARPs with damaged DNA at first reduces the access of other DNA-binding proteins, including pol β. The subsequent activation of PARPs leads to the formation of PAR and to the autoPARylation of the proteins themselves; a consequence is the departure of the PARP and the release of the DNA for the implementation of pol β activity. On the other hand, PAR formation has potentially a dual effect on pol β because this enzyme could be recruited to DNA damage sites via PAR binding but simultaneously undergoes PARylation, which weakens affinity for negatively charged DNA [20]. Supplementation with HPF1 dramatically changes PAR targets: a substantial switch of PARylation from PARPs and pol β to core histones. On the other hand, this phenomenon leads to NCP relaxation and consequently gives pol β access to DNA. On the other hand, the magnitude of the positive effect should be highly dependent on the PARylation stage.

A recent publication indicates that HPF1 is not a necessary component for the maintenance of genome integrity in the presence of single-strand-DNA break-induced agents but still shows that the absence of this protein significantly influences the efficiency of BER in a PARP1-dependent manner [44]. The influence of HPF1 may be considerable during single damage in easily accessible regions of the NCP; alternatively, HPF1 has to be part of a complex with PARP2 to resolve such situations. For lesions located in hard-to-reach regions, which require substantial alterations to the chromatin structure, the ATP-dependent chromatin remodeling system is required [10,45]. In this case, the contribution of histone PARylation on serine becomes less significant.

## Figures and Tables

**Figure 1 ijms-26-01794-f001:**
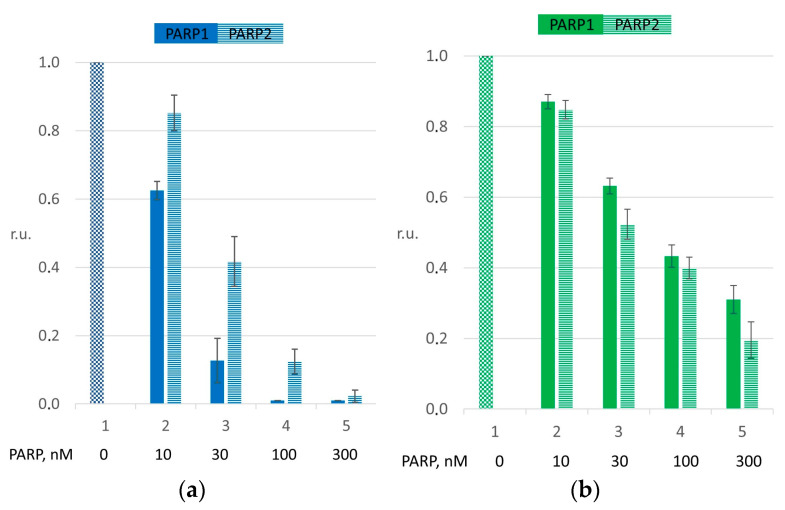
The suppression of single-nucleotide gap filling catalyzed by pol β caused by PARP1 or PARP2. The histograms reflect dTMP incorporation efficiency with naked DNA (**a**) or the NCP (**b**). The reaction mixtures contained 10–300 nM PARPs (bars 2–5). The magnitude is presented in relative units, normalized to the of pol β efficiency seen when dTMP was incorporated without the addition of PARP (bar 1). The data are presented as averages of at least three independent experiments and are shown as means ± SDs. The corresponding *p*-values are in the main text.

**Figure 2 ijms-26-01794-f002:**
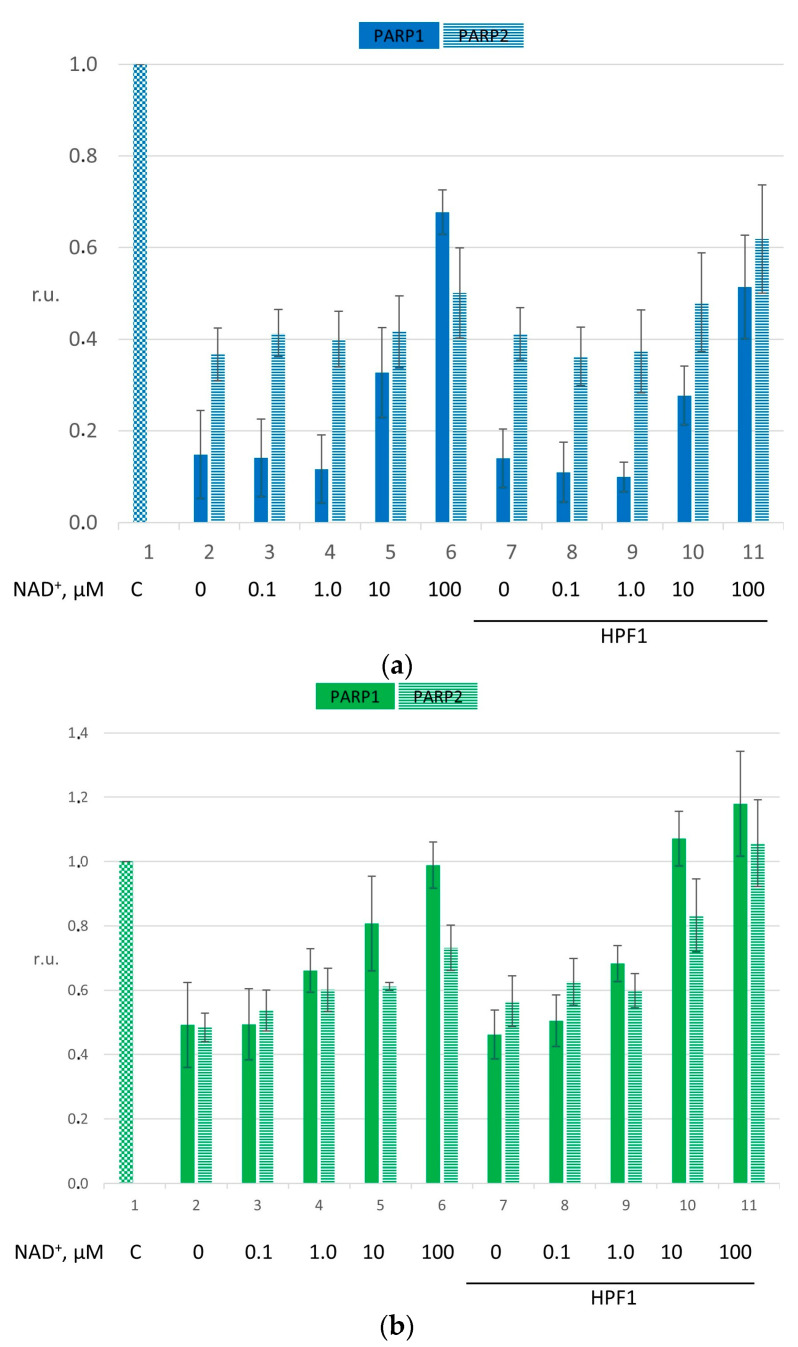
The impact of HPF1 on single-nucleotide gap filling under PARP1- or PARP2-driven PARylation. The efficiency of nucleotide incorporation in gapped DNA in the presence of either naked DNA (**a**) or the NCP (**b**). The concentrations of PARPs were chosen to be 30 and 100 nM for naked DNA and the NCP, respectively (bars 2 and 7). The concentrations of HPF1 were chosen so that they were in a 2:1 molar ratio with respect to the PARP concentrations, i.e., 60 nM for the experiments with naked DNA and 200 nM for the experiments with the NCP. The concentration of NAD^+^ was varied in the range from 0.1 to 100.0 μM (bars 3–6 and 8–11). The magnitudes are presented in relative units, r.u., normalized to the pol β efficiency seen when dTMP was incorporated without additional factors such as PARP1, PARP2, HPF1 and NAD^+^ (bar 1). The data are presented as averages of at least three independent experiments and are shown as means ± SDs. The corresponding *p*-values are in the main text.

**Figure 3 ijms-26-01794-f003:**
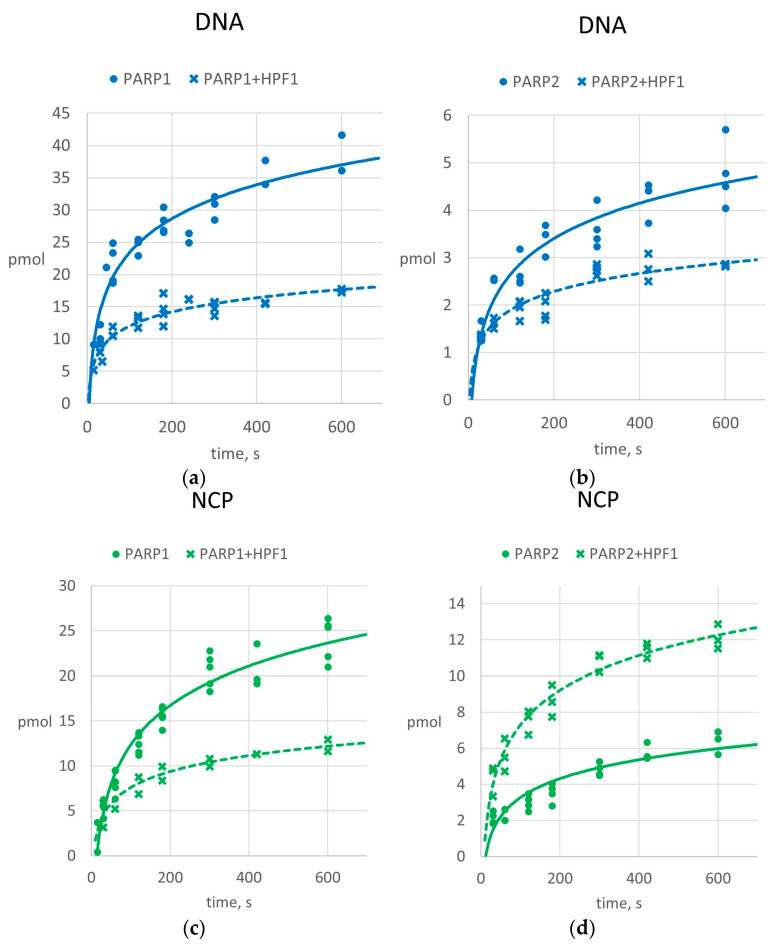
HPF1 affects PAR formation by PARP1 (**a**,**c**) and PARP2 (**b**,**d**) in a single-nucleotide gap-filling reaction. The solid curves represent the kinetics of PAR accumulation with a PARP alone, and the dashed curves correspond to the reactions in the presence of HPF1. The respective raw data points are marked with circles and crosses. The X-axis shows time, in seconds; the Y-axis denotes PAR formation efficiency, in pmol (see Materials and Methods). The *R^2^* values for the data obtained in the absence or in the presence of HPF1, are as follows: (**a**) 0.88 or 0.87, (**b**) 0.85 or 0.86, (**c**) 0.94 or 0.89 and (**d**) 0.86 or 0.94, respectively. The quantitative data were analyzed as described in the Methods section and are presented along with the coefficient of determination *R*^2^.

**Figure 4 ijms-26-01794-f004:**
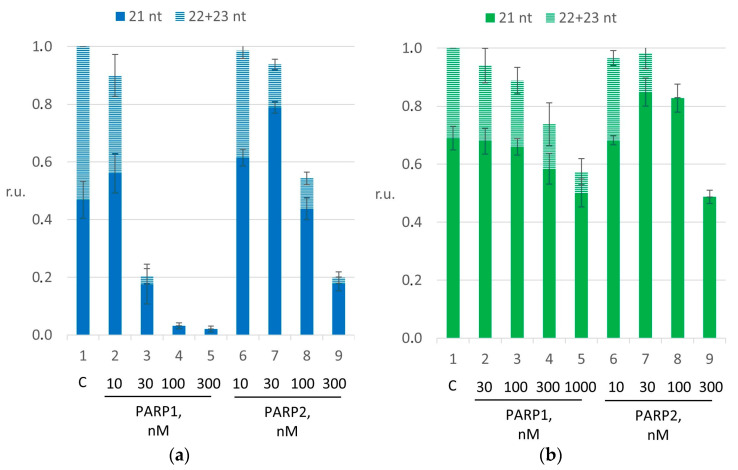
The suppression of DNA strand displacement caused by PARP1 or PARP2. Panel (**a**) presents experimental results during DNA strand-displacement synthesis on naked DNA; panel (**b**) presents DNA strand-displacement synthesis on the NCP. The magnitudes are presented in relative units, normalized to the initial concentration of DNA or NCP. The reaction mixtures contained 10 nM naked DNA, 3.0 nM pol β, 100 µM each dNTP and 10–300 nM PARP1 or PARP2. Other reaction mixtures contained 10 nM NCP, 1.0 µM pol β, 400 μM four dNTPs and 30–1000 nM PARP1 or PARP2. The data are presented as averages of at least three independent experiments and are shown as means ± SDs. The corresponding *p*-values are in the main text.

**Figure 5 ijms-26-01794-f005:**
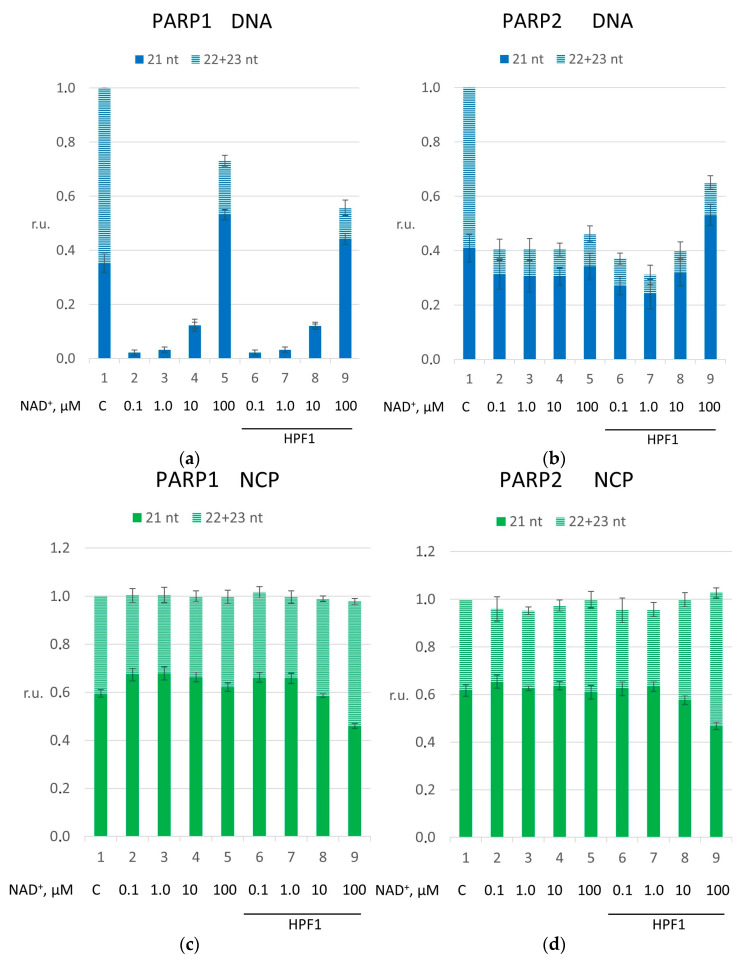
The impact of HPF1 on the strand-displacement synthesis catalyzed by pol β under PARylation carried out by PARP1 (**a**,**c**) or PARP2 (**b**,**d**). The efficiency of nucleotide incorporation into either naked DNA (panels (**a**,**b**)) or the NCP (panels (**c**,**d**)). The dark part of the bar represents the incorporation of the first-filling nucleotide; the light part of the bar denotes the incorporation of subsequent nucleotides accompanied by DNA strand displacement. The concentration of NAD^+^ was varied in the range from 0 to 100 µM. The magnitude is presented in relative units, normalized to the initial DNA or NCP concentration. The data are presented as averages of at least three independent experiments and are shown as means ± SDs. The corresponding *p*-values are in the main text.

**Figure 6 ijms-26-01794-f006:**
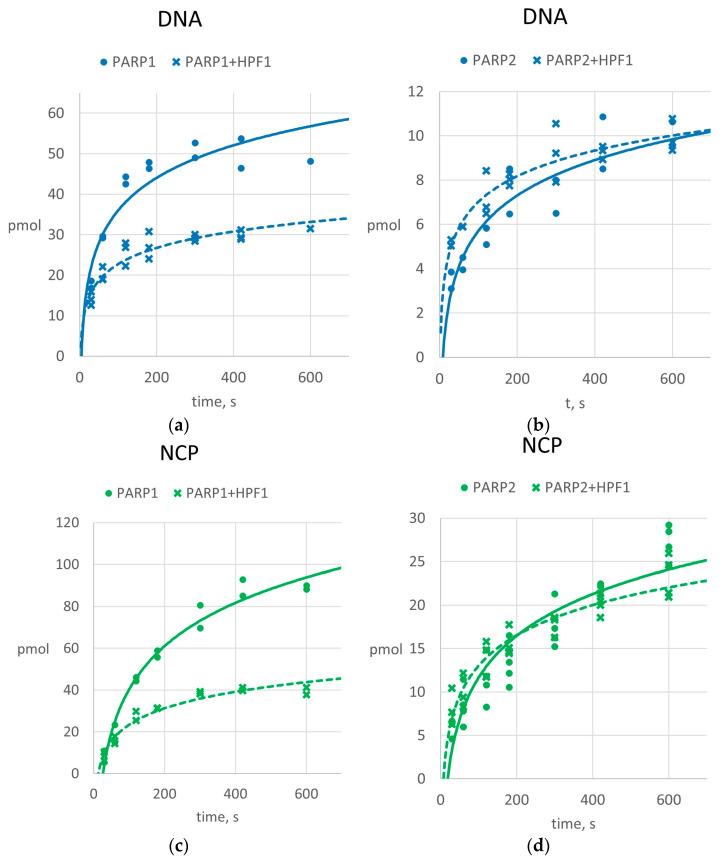
The impact of HPF1 on PAR formation under DNA strand-displacement conditions. The solid curves denote the kinetics of PAR accumulation caused by a PARP alone, and the dashed curves represent the reactions in the presence of HPF1. The raw data points are marked with circles and crosses. The X-axis represents time, in seconds; the Y-axis denotes PAR formation efficiency, in relative units (see Materials and Methods). The *R*^2^ values for the data obtained in the absence or in the presence of HPF1 are as follows: (**a**) 0.86 or 0.87, (**b**) 0.86 or 0.86, (**c**) 0.97 or 0.94 and (**d**) 0.85 or 0.89, respectively. The quantitative data were analyzed as described in the Methods section and are presented together with the coefficient of determination *R*^2^.

## Data Availability

Data are contained within the article and Appendix A.

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
