# Peer review of "HPF1 Regulates Pol β Efficiency in Nucleosomes via the Modulation of Total Poly(ADP-Ribose) Synthesis"

_ijms, 2025, doi:10.3390/ijms26051794_

Round 1
Reviewer 1 Report
Comments and Suggestions for Authors
In this manuscript, Kutuzov et al. studied about the role of HPF1 in modulating polβ activity during the repair of gap-containing DNA mediated by PARP1/2. PARylation is required for efficient gap-filling and strand-displacement synthesis by polβ. HPF1 regulates PARP activity, increasing in autoPARylation of PARPs and histone PARylation, which facilitates polβ activity by dissociating DNA from PARPs and core histones.
1. Line 150-157 speculates that PARP1/2 binds to DNA substrate and blocks polβ activity. However, the direct evidence is not shown in this manuscript. I think such information should be included in introduction or discussion.
2. In Figure 1-4, it is difficult to interpret the difference of binding affinity from bar graph. Is it possible to estimate IC50 from the experiments?
3. Statistical test should be applied to all applicable data.
Comments on the Quality of English LanguageIt is helpful if the interpretation of the experimental results is described more clearly.
Author Response
Comments 1: Line 150-157 speculates that PARP1/2 binds to DNA substrate and blocks polβ activity. However, the direct evidence is not shown in this manuscript. I think such information should be included in introduction or discussion.
Response 1: We agree, but we would like to keep a modified version of this statement in the Results and Discussion, with your permission. The Results are combined with Discussion in our paper. This paragraph deals with a discussion of the data presented in Fig. 1 and Fig. 2(A) and 2(B), bars 1 and 2. The raw data, from which the quantitative data of Figs. 1 and 2 were calculated, are presented in Fig. S2. The process in question involves the incorporation of a single nucleotide by DNA polymerase beta into a partial DNA duplex with a gap containing 3'-OH and 5'-dRp moieties. Two reactions are examined: one involving naked DNA and the other involving DNA within a nucleosome. Both structures are substrates for DNA polymerase beta as well as for PARP1 and PARP2. In this case, substrate specificity of DNA polymerase beta follows from the fact that in the presence of an appropriate dNTP, the DNA synthesis proceeds. As for the proteins PARP1 and PARP2, indeed, we did not conduct the experiments to determine the dissociation constant of PARP complexes with DNA and nucleosomes within the framework of the current study. However, these data are widely reported in the literature. In particular, in our laboratory, the Kd values were obtained for the corresponding complexes by the gel shift assay [Ukraintsev, A.A.; Belousova, E.A.; Kutuzov, M.M.; Lavrik, O.I. Study of Interaction of the PARP Family DNA-Dependent Proteins with Nucleosomes Containing DNA Intermediates of the Initial Stages of BER Process. Biochemistry (Mosc) 2022, 87(4), pp. 331-345; here, it’s ref. 28] and by the fluorescence anisotropy method [Kutuzov, M.M.; Belousova, E.A.; Kurgina, T.A.; Ukraintsev, A.A.; Vasil'eva, I.A.; Khodyreva, S.N.; Lavrik O.I. The contribution of PARP1, PARP2 and poly(ADP-ribosyl)ation to base excision repair in the nucleosomal context. Sci Rep 2021, 11(1), pp. 484944; here it’s ref. 20]. In general, the difference in the catalytic activity between these proteins was shown in ref. [Amé, J.C.; Rolli, V.; Schreiber, V.; Niedergang, C.; Apiou, F.; Decker, P.; Muller, S.; Höger, T.; Ménissier-de Murcia, J.; de Murcia, G. PARP-2, A novel mammalian DNA damage-dependent poly(ADP-ribose) polymerase. J Biol Chem 1999, 274(25), pp. 17860-8; here, it’s ref. 16], and the difference in the affinity and substrate specificity between PARP1 and PARP2 for the same substrate is described in ref. [Crystal structure of the catalytic fragment of murine poly(ADP‐ribose) polymerase‐2 Antony W. Oliver, Jean‐Christophe Amé, S. Mark Roe, Valerie Good, Gilbert de Murcia, Laurence H. Pearl Nucleic Acids Research, Volume 32, Issue 2, 16 January 2004, Pages 456–464, https://doi.org/10.1093/nar/gkh215]. On the basis of the presented data, we discuss the competition for the substrate between DNA polymerase beta and each PARP protein in the interaction.
We have inserted the references to these literary sources into the Introduction section and into the part of the Results and Discussion section mentioned in the comment. We also updated this part of the text accordingly.
Comment 2: In Figure 1-4, it is difficult to interpret the difference of binding affinity from bar graph. Is it possible to estimate IC50 from the experiments?
Response 2: If we understand the remark correctly, the Reviewer is talking about the inhibition of DNA synthesis catalyzed by DNA polymerase beta in the presence of PARP1 or PARP2. As far as we know, IC50 is used as a measure of the potency of a substance in inhibiting a specific reaction and is not common for characterization of protein–protein interactions. Here, we talking about the process of inhibition, i.e., a decrease, of the DNA synthetic activity of pol beta by a protein: PARP1 or PARP2. In other words, we didn’t use any inhibitors, for instance small-molecule compounds, that could affect the DNA polymerase activity.
In actuality, the histograms in Figs. 1, 2, 4, and 5 show the yield of the polymerase reaction catalyzed by DNA pol beta in the presence of PARP1 or PARP2 either alone or in combination with HPF1. Definitely, it is dependent on the affinity of PARPs for DNA or for the NCP. Here, we aimed to perform only a functional analysis because the affinity data are widely available in the literature. In our previous studies [20, 28], we have evaluated the affinity of PARP1 and PARP2 for naked gapped DNA and for an NCP containing a gap in the region of the DNA helix accessible for protein interaction (i.e., the gap was formed after excision of the “out”-oriented nucleotide). The measurement methods were the gel shift assay (EMSA) and fluorescence anisotropy (an. fluor.) The Kd values were as follows: Kd [PARP1, gap-DNA] = ~29 nM (an. fluor.), Kd [PARP1, gap-NCP] = ~10 nM (an. fluor.) and ~38 nM (EMSA); Kd [PARP2, gap-DNA] = ~58 nM (an. fluor.), Kd [PARP2, gap-NCP] = ~14 nM (an. fluor.) and ~57 nM (EMSA).
We have made the appropriate edits in the text, page 4.
Comments 3: Statistical test should be applied to all applicable data
Response 3: Thank to Reviewer for this comment. We have inserted the requested data into the text in the Methods section as well as into figure legends.
Reviewer 2 Report
Comments and Suggestions for Authors
The authors present a manuscript about the relationship between HPF1 and pol b and how this affects the efficiency of the enzyme synthesizing DNA in the context of the BER system's action. The authors show many kinetic experiments. However, some aspects need improvement.
Major Issues:
1. The plots presented in this manuscript show the activity of pol b and how this activity is modified with several factors. However, when the authors say in the main text that the enzyme's activity has changed, no significant difference is observed in the figures. It is necessary to add a statistical test to support all these kinds of affirmations in the text.
2. In Figure 2, panel B shows experimental points with higher RU than the control (RU = 1). Could you explain this result?
3. Lane 187 - 188: The authors indicate in the text that "there are significant differences." However, no statistical tests support this point. Please correct this.
4. The authors perform experiments using pol b protein. However, there are no references to this protein in the methods section. For example, is this protein recombinant? Was it purified in the laboratory?. This point is valid with the other enzymes used in the enzymatic assays.
5. The weak point in this work is the statistics. A new paragraph in the methods section should indicate the statistical tests used to analyze data. In addition, it is necessary to add the protocol, data analysis, and quantification method to generate the graphics presented. In other words, the authors took the results shown in supplementary figures and transformed them into the graphics presented in the main text.
Minor Issues:
1. The abstract must be corrected because there are no references to the enzyme pol b, which is an essential part of the work.
2. The introduction section does not mention the enzyme pol b. Some lanes containing relevant information about the enzyme need to be added.
Author Response
Comments 1: The plots presented in this manuscript show the activity of pol b and how this activity is modified with several factors. However, when the authors say in the main text that the enzyme's activity has changed, no significant difference is observed in the figures. It is necessary to add a statistical test to support all these kinds of affirmations in the text.
Response 1: Thank to Reviewer for this comment.
In this work, we performed in vitro experiments with a limited set of proteins that convert a specific substrate. Here, the dispersion is largely determined by the experimental error and sensitivity of the method rather than by variation of values of the enzyme's activity. For this type of data, the presentation of error bars typically provides comprehensive information about the reliability of the measurement. For this reason, to estimate the dispersion of the data, we used standard deviation calculated from three independent experiments. These SD values are shown in the graphs. In addition, in this work, we are more interested in the trends that appear as the effect of a particular factor (for example, PARP1 and PARP1+HPF1) than in the absolute changes of the parameter.
Probably the information about calculations and statistical test were presented unclearly; therefore, we added the respective data into the Methods section and into figures legends.
Comments 2: In Figure 2, panel B shows experimental points with higher RU than the control (RU = 1). Could you explain this result?
Response 2: Sorry, for the lack of clarity. In this work, we evaluated the effect of PARP proteins on the polymerase activity of DNA polymerase beta. Figures 1, 2, 4, and 5 show the levels of DNA substrate extension in the presence of either PARP1/2 or PARP1/2 with HPF1. The level of the substrate extension in the absence of any factors was taken as one unit. Accordingly, with the inhibitory effect of the protein and a decrease in the level of DNA synthesis, a bar was less than 1, and if the presence of protein factors leads to the activation of DNA polymerase, then a bar was greater than one. For example, for bar 11 in Fig. 2B, this means that in this case, the efficiency of dTMP incorporation catalyzed by DNA polymerase beta in the presence of PARP1 and HPF1 under conditions of active PARylation is higher than the same parameter for DNA polymerase beta in the absence of any factors.
We have included a corresponding explanation in the legend to Fig. 2.
Comments 3: Lane 187 - 188: The authors indicate in the text that "there are significant differences." However, no statistical tests support this point. Please correct this.
Response 3: There probably was some misunderstanding related to the wording. Here, we are not talking about a quantitative statistical evaluation, it is just a figure of speech. We have replaced the word “significant” here and in several other places.
Comments 4: The authors perform experiments using pol b protein. However, there are no references to this protein in the methods section. For example, is this protein recombinant? Was it purified in the laboratory?. This point is valid with the other enzymes used in the enzymatic assays.
Response 4: Apologies for not mentioning this information in Methods subsection (4.2), however the description of origins of the proteins are presented in the Materials subsection (4.1). In this work, all the enzymes were recombinant: human APE1, rat DNA polymerase beta (pol b), human PARP1, murine PARP2, T. aquaticus Taq DNA polymerase, and E. coli uracil-DNA glycosylase. The Histone PARylation Factor 1 from human was also recombinant. Histones H2A, H2B, H3, and H4 were purified from the erythrocytes of G. gallus.
Comments 5: The weak point in this work is the statistics. A new paragraph in the methods section should indicate the statistical tests used to analyze data. In addition, it is necessary to add the protocol, data analysis, and quantification method to generate the graphics presented. In other words, the authors took the results shown in supplementary figures and transformed them into the graphics presented in the main text.
Response 5: Sorry for not mentioning these details. As suggested, we have updated methods and figure legends. In all the figures, the data are presented as an average of at least three independent experiments and are shown as the mean ± SD. The gels with the separated reaction products were visualized with a Typhoon FLA 9500 scanner and the yield was analyzed in the Quantity One software (Bio-Rad, USA). The quantitative data were analyzed in Microsoft Excel 2010 and presented in histograms as the mean ± SD.
Minor Issues:
- The abstract must be corrected because there are no references to the enzyme pol b, which is an essential part of the work.
- The introduction section does not mention the enzyme pol b. Some lanes containing relevant information about the enzyme need to be added.
Sorry for this oversight, we have inserted the relevant information into the aforementioned parts of the text.
Round 2
Reviewer 1 Report
Comments and Suggestions for Authors
The authors just added the description of bar plots (Figure 1, 2, 4, 5) or the square of the correlation coefficients (Figure 3, 6). The statistical test is still needed if you mention the experimental conditions caused the difference of the results.
Comments on the Quality of English LanguageIt is helpful if the interpretation of the experimental results is described more clearly.
Author Response
The authors just added the description of bar plots (Figure 1, 2, 4, 5) or the square of the correlation coefficients (Figure 3, 6). The statistical test is still needed if you mention the experimental conditions caused the difference of the results.
We have inserted the requested data into the main text and into Methods section.
The revised version of the manuscript is an improvement over the submitted version by English editing service.
Reviewer 2 Report
Comments and Suggestions for Authors
Thank you to the authors for the improvement of the manuscript.
Author Response
We are grateful to the Reviewer for his comment.